# Stem Cell Therapy for Diabetes: Beta Cells versus Pancreatic Progenitors

**DOI:** 10.3390/cells9020283

**Published:** 2020-01-23

**Authors:** Bushra Memon, Essam M. Abdelalim

**Affiliations:** 1College of Health and Life Sciences, Hamad Bin Khalifa University (HBKU), Qatar Foundation, Education City, P.O。 Box 34110 Doha, Qatar; bmemon@mail.hbku.edu.qa; 2Diabetes Research Center, Qatar Biomedical Research Institute (QBRI), Hamad Bin Khalifa University (HBKU), Qatar Foundation (QF), P.O. Box 34110 Doha, Qatar

**Keywords:** hPSCs, hyperglycemia, insulin-secreting cells, β-cell precursors, pancreatic islets, transplantation

## Abstract

Diabetes mellitus (DM) is one of the most prevalent metabolic disorders. In order to replace the function of the destroyed pancreatic beta cells in diabetes, islet transplantation is the most widely practiced treatment. However, it has several limitations. As an alternative approach, human pluripotent stem cells (hPSCs) can provide an unlimited source of pancreatic cells that have the ability to secrete insulin in response to a high blood glucose level. However, the determination of the appropriate pancreatic lineage candidate for the purpose of cell therapy for the treatment of diabetes is still debated. While hPSC-derived beta cells are perceived as the ultimate candidate, their efficiency needs further improvement in order to obtain a sufficient number of glucose responsive beta cells for transplantation therapy. On the other hand, hPSC-derived pancreatic progenitors can be efficiently generated in vitro and can further mature into glucose responsive beta cells in vivo after transplantation. Herein, we discuss the advantages and predicted challenges associated with the use of each of the two pancreatic lineage products for diabetes cell therapy. Furthermore, we address the co-generation of functionally relevant islet cell subpopulations and structural properties contributing to the glucose responsiveness of beta cells, as well as the available encapsulation technology for these cells.

## 1. Introduction

Diabetes mellitus occurs due to the loss or impaired function of insulin-secreting pancreatic beta cells. There are two main types of diabetes, including type 1 (T1D) and type 2 (T2D). T1D is characterized by beta cell destruction as a result of autoimmune defect, while T2D pathogenesis involves the development of insulin resistance in the insulin-target tissues followed by beta cell dysfunction due to a combination of genetic and environmental factors [1]. Monogenic diabetes (MD), a rare form of the disease, is characterized by genetic mutations in a specific gene involved in pancreatic development and/or beta cell function [2]. Recent therapeutic approaches aim to restore the endogenous production of insulin rather than the conventional insulin injection treatment. Allogenic islet transplantation is the cell therapy option in practice for the management of T1D. However, this approach is limited by the requirement of an adequate number of functional islets for achieving normal glycemic levels from cadaveric pancreas. Additionally, the long-term necessity for immuno-suppressants has hindered its application at early stages of T1D as well as in children. The survival, proliferation, and functionality of isolated islets following transplantation is also an unresolved complication [3]. Therefore, other beta cell replacement options are being evaluated for clinical applications [4].

Human pluripotent stem cells (hPSCs) serve as a valuable tool for cell therapy and disease modeling. Further, hPSCs can be classified into two types-embryonic stem cells (hESCs) that are derived from the inner cell mass of the embryo and induced PSCs (hiPSCs) that are generated by somatic cell reprogramming. hPSCs can be expanded in vitro to provide an unlimited starting source for beta cell generation and can be differentiated into any cell type under appropriate cues [5,6]. hPSC-derived pancreatic lineage products, with the advantage of being scalable and personalized, are a promising alternative for cell therapy [5]. While step-wise protocols for the differentiation of hPSC-derived pancreatic progenitor cells that give rise to mono-hormonal insulin-secreting cells within the body have been designed, the in vitro generation of functional pancreatic beta cells has been challenging [7] (Figure 1 and Figure 2). To date, only one clinical trial has been approved for hPSC-derived product for T1D treatment, led by the company ViaCyte (ClinicalTrials.gov Identifier: NCT03163511, NCT02239354). hPSC-derived pancreatic progenitors, that bear the potential to mature into insulin-secreting cells in vivo, were encapsulated in an immune-isolation device and transplanted subcutaneously in T1D patients (Figure 1). While results of the trial evaluating the safety and efficacy of the candidate product are much awaited, it is worthwhile to discuss what pancreatic-lineage product would be a better contender as the diabetes cell therapy candidate, the terminally differentiated functional beta cells or their precursors, the multipotent pancreatic progenitors. This leads to the discussion of which candidate would provide a more robust physiological control over glycemic levels, similar to that of a healthy individual. To determine an ideal cell therapy candidate, multiple factors such as the convenience of the method to generate it in vitro for scalability, purity and composition of the cell population being encapsulated, functionality of the cells, and the ability to control hyperglycemia should be critically assessed. These conditions for hPSC-derived pancreatic lineage products are discussed below.

## 2. Generation of Pancreatic Progenitors and Beta Cells from Human Pluripotent Stem Cells (hPSCs)

Pancreatic and duodenal homeobox 1 (PDX1) and homeobox protein NKX6.1 (NKX6.1) co-expressing cells within the developing human embryo mark the multipotent pancreatic bud and trunk progenitors that later generate insulin-secreting beta cells [7]. Both key transcription factors (TFs) are highly expressed in pancreatic progenitor cells and functional insulin-secreting beta cells. It has been shown that the co-expression of PDX1 and NKX6.1 is required for the generation of mono-hormonal, glucose-responsive beta cells [9,10]. Specifically, NKX6.1 is a crucial marker that regulates beta cell maturation and functionality [9,11]. Typically, hPSCs are differentiated through different stages mimicking embryonic development, such as definitive endoderm and posterior foregut sequentially to yield pancreatic and endocrine progenitors, with the expression of a specific set of regulatory proteins at each stage (Figure 1 and Figure 3). A highly efficient definitive endoderm is induced by Activin/Nodal signaling [12]. The activation of FGF and retinoid signaling, along with the inhibition of BMP and hedgehog signaling, directs the endodermal cells to a pancreatic fate [13,14]. EGF and nicotinamide signaling enhance the generation of pancreatic progenitors and are crucial for the induction of NKX6.1 expression [15]. The inhibition of Notch signaling following the generation of PDX1+/NKX6.1+ progenitors is required for the efficient specification of Neurogenin (NGN3)+ endocrine progenitors [16]. Recent protocols have incorporated TGF beta inhibitors to enhance endocrine cell differentiation [8,17,18,19]. Therefore, in vitro differentiation strategies employ a combination of small molecules or recombinant proteins to turn on or off these pathways at specific stages of the protocol [7,20]. 

Multiple groups have differentiated hPSCs to PDX1 and NKX6.1 co-expressing pancreatic progenitors in monolayer cultures [15,22], yielding even up to 90% of PDX1+/NKX6.1+ co-positive pancreatic progenitors [21,23] (Figure 3). Prior to the most recent improvements in protocols employing the monolayer method, studies have shown that cellular aggregation during the differentiation process could improve the efficiency of pancreatic progenitors as compared to monolayer cultures [17,24]. Culturing cells as aggregates or embryoid bodies in suspension, particularly from the endocrine progenitor stage, improves the efficiency and functionality of the generated beta cells [18,19,25]. Several reports have shown the generation of pancreatic beta cells that express INSULIN (INS) but are not functional [26,27]. Those non-functional beta cells usually express INS, which is co-localized with GLUCAGON (GCG) and/or SOMATOSTATIN (SST) (polyhormonal cells) [28]. Also, they do not express key beta cell markers, such as MAFA, NKX6.1, or transporters and channels that aid in glucose-stimulated INS secretion. However, mature functional pancreatic beta cells co-express INS with NKX6.1 and MAFA and secrete C-peptide in response to elevated glucose levels with the help of the cell’s secretory machinery [8,18,19]. 

In vitro differentiation of hPSCs to pancreatic progenitors takes two weeks and is less variable in terms of the efficiency of generating PDX1 and NKX6.1 co-positive pancreatic progenitors [9,15,21] with highly efficient protocols being recently established (~80–90% PDX1+/NKX6.1+ cells) [15,21] (Figure 3). However, the differentiation of hPSCs to insulin-secreting beta cells is longer and may take more than a month to generate mature beta cells (Table 1). One of the main challenges is the variability in the differentiation efficiencies of different hPSC lines. A previous study reported that applying the same pancreatic progenitor protocol on eight different hPSC lines results in a variation in NKX6.1 induction ranged from 37% to 84% [15]. Also, in our lab, applying the most recent differentiation protocols for beta cells [8,19] and pancreatic progenitors [15,21] on several hPSC lines (hESCs/hiPSCs) showed clear variations in the differentiation efficiencies from one cell line to another cultured under the same conditions (unpublished data). These findings demonstrate that the differentiation of pancreatic progenitors as well as beta cells is dependent on the inherent differences across cell lines as well as on the robustness of the differentiation protocols. Although recent protocols showed the generation of glucose-responsive beta cells in vitro [8,17,18,19], their functionality need to be improved to be suitable for transplantation therapy. Nevertheless, in vivo maturation of hPSC-derived pancreatic progenitors and beta cells have both successfully reversed hyperglycemia in mice, with pancreatic progenitor cells taking longer than beta cells for achieving euglycemia due to the prolonged maturation period in vivo [8,17,18,29,30]. While the viability, and hence, developmental capacity of hPSC-derived pancreatic progenitor, immune-protected during the longer maturation period in vivo has been demonstrated in rodents [9,30,31], it is yet to be determined in humans. The detailed results from the first phase of the clinical trial by ViaCyte are expected to shed light on this.

Substantial progress has been achieved in the differentiation of hPSCs into pancreatic beta cells. The efficiency of generating beta cells in vitro with the recent protocols has improved to a ~40% for NKX6.1^+^/INS^+^ or C-PEPTIDE^+^ cells [8,19]. However, these beta cells, although glucose-responsive, are still dissimilar to adult human beta cells in other aspects of functionality, such as lower expression levels of GLIS3, PCSK2 [19], PAX6, and KCNK3 [8] as well as a reduced calcium response on glucose induction compared to adult islets [8]. A recent study showed that the reclustering of immature INS+ cells generates hPSC-derived beta cells close to adult human beta cells. However, those beta cells express low levels of maturity markers UCN3, MAFA, and G6PC2 in comparision to adult human beta cells [18]. Most of the differentiation protocols aim to get rid of the polyhormonal cells. However, it has been found that a subset of islet cells, especially in the fetal pancreas, are polyhormonal [40]. These polyhormonal cells that co-express INS with other hormones have been found to be particularly increased in lean T2D individuals [41], suggesting that they may possess an undetermined role in maintaining euglycemia and adapting to stress. Previous attempts at characterizing these polyhormonal endocrine cells demonstrated that they secrete INS and GCG upon depolarization, but not in response to glucose [28], and they may specify into alpha (GCG+) cells in vivo [9,33,42]. These alpha cells may play a role in the postnatal maturation of beta cell function. Therefore, it is worth studying if the transplantation of a common progenitor population, rather than terminally differentiated individual beta cells, that would differentiate in vivo into all relevant endocrine populations regulating beta cell function and maturation, could be the key to achieving euglycemia. 

It is also plausible that there exist multiple progenitor populations that could give rise to beta cells during different stages. Previous studies have reported other pancreatic lineages that may be used to generate insulin-secreting cells for diabetes treatment. Recently, Aigha et al. generated a novel population of pancreatic progenitor population expressing NKX6.1 in the absence of PDX1 (PDX1-/NKX6.1+) from hPSCs. The initial characterization of this population indicates the potential of PDX1-/NKX6.1+ cells to generate islet endocrine cells [23]. In vitro expanded islets undergoing de-differentiation have been shown to generate INS+ beta cells on extended culture [43,44], as well as progenitor cells with high aldehyde dehydrogenase activity extracted from the fetal and adult pancreas [45,46]. Additionally, the pancreatic duct serves as a reservoir of a subset of progenitor cells possessing ability to transdifferentiate to beta cells [47,48]. For example, a cell population expressing P2RY1 and ALK3 (P2RY1+/ALK3+) has been found in the pancreatic ducts and ductal glands. This cell population expresses PDX1 and NKX6.1, indicating the existence of a new source of insulin secreting cells in the pancreatic ducts [49]. In addition, pancreatic exocrine tissue and other pancreatic endocrine cell types have also been shown to transdifferentiate to INS+ cells [48,50,51,52]. There is no clear consensus on what these progenitors of INS+ cells represent. However, they have been broadly termed as pancreatic stem cells, with no defined criteria for the expression of specific markers, lineage of origin, or location within the pancreas [53]. However, their differentiation efficiency in generating insulin-secreting cells in vitro is yet to be fully gauged, and thus, their potential of serving as a cell therapy candidate remains unexplored. 

## 3. Isolation of Pure Human Pluripotent Stem Cells (hPSC)-Derived Pancreatic Lineages

While hPSC-derived pancreatic progenitors could be associated with the co-generation of mesodermal and ductal lineages, the generated hPSC-derived functional beta cells are contaminated by non-committed progenitor cells and polyhormonal cells that are non-functional, as well as other cell types. Therefore, in order to be able to employ either of the two candidates for cell therapy, it is critical to isolate and transplant a homogenous population. For this purpose, focus has been directed towards the identification of cell surface markers distinguishing pure beta cells or pancreatic progenitors committed towards beta cell lineage. The identification of markers such as CD24, CD142, and Glycoprotein 2 (GP2) as defining markers for beta-cell precursors has been proposed [22,32,33,34,36]. For characterizing endocrine progenitor population, CD133 and CD49f were identified to distinguish Neurogenin 3 (NGN3)-expressing cells during human and mouse development [54]. Additionally, SUSD2, CD200, and CD318 have been shown to identify endocrine progenitors (NGN3+ cells) in the human pancreas and hPSC-derived endocrine pancreatic progenitors [33]. Recently, Melton’s team identified CD49a (ITGA1) as a surface marker expressed on pancreatic beta cells [35]. Since in vitro hPSC-derived beta cell protocols do not yield a pure population of beta cells, CD49a could be used for isolating the small proportion of functional beta cells generated within the pancreatic organoid in vitro using these protocols, and re-aggregate pure beta cells prior to encapsulation for transplantation. However, fractionating the differentiated pancreatic organoids into individual cells for the purpose of isolating only beta cells would disrupt the islet architecture generated in the dish that is sought to recapitulate in vivo development. To this extent, it has been demonstrated that isolated pancreatic beta cells underperform those in intact islets in insulin secretion response [55,56,57], indicating that establishment of the strategic cell-cell and cell-matrix contacts by beta cells with the islet is rather crucial to conferring its functional properties. In contrast, the isolation of pure beta cells generated within the hPSC-derived pancreatic organoid and their re-aggregation to establish beta cell–beta cell contact improved their metabolic maturation and glucose-responsiveness [18]. Interestingly, a combination of novel surface markers, Hpi2^+^/HPα1^−^/HPx2^−^, as well as CD9^+^/CD56^+^, used to enrich pancreatic beta cell fraction from whole pancreas, co-purified a small fraction of delta cells [58,59]. Complementing this finding, NKX6.1^high^ progenitors also matured into SST-positive cells, in addition to beta cells [9]. This highlights that certain endocrine cells may have a common stem cell progenitor and hence may also share similar cell surface proteome. Hence, the identification of a cell surface marker exclusive to beta cells would require extensive characterization. This underscores the need for deciphering how other endocrine cells assist in the maturation of pancreatic beta cell function, and if they should be part of the ideal cell therapy candidate. 

## 4. Intra-Islet Cell Contact and Communication for Functional Performance

Pancreatic islet architecture with the appropriate distribution of different endocrine cell types has been shown to contribute towards beta cell functionality and glucose responsiveness. Resonating from the findings that beta cells within an intact islet are more precise in their secretory and calcium response to elevated glucose levels than those in dissociated or dispersed islets [55,56,57], it is important to consider that intra-islet cell-cell communication between beta cells and other endocrine cells, as well as amongst different subpopulations of beta cells, is crucial for paracrine signaling and enhanced electrical coupling [60,61,62].

### 4.1. Role of Pancreatic Endocrine Cells in Reversal of Hyperglycemia

The presence of GCG-secreting alpha cells and SST-secreting delta cells have been shown to affect insulin secretion, highlighting the critical role of intra-islet paracrine signaling by the endocrine cells, in addition to that of the nervous and vascular system [62,63,64,65,66]. It is crucial to note that these secreted hormones are present at a higher concentration within the islet, thereby making the signaling cascades downstream of their receptors impactful in the islet endocrine cells. GCG and GCG-derived peptides have been shown to positively regulate the exocytosis of insulin-containing vesicles [67,68]. In addition to the paracrine inhibitory effect of GCG and SST on insulin secretion, it has been found that alpha cells affect beta cell function through other secretagogues like acetylcholine [65]. On the other hand, SST secreted from the delta cells of the islet inhibits insulin secretion [69]. In addition, endocrine cells of the islet, such as alpha cells have been shown to have a high expression of insulin receptor (INSR). Furthermore, excessive insulin has been shown to have paracrine inhibitory effects on GCG secretion [70]. Therefore, it is worthwhile to note that, due to the loss of insulin signaling from the compromised beta cells in diabetes mellitus, the development and functionality of the other endocrine cells could be affected [71]. In this regard, it has been proposed that, in the T2D pancreas, there is an increase in the proportion of alpha to beta cells due to diminished beta cell numbers [72]. Also, a recent study demonstrated that pancreatic alpha cells are more resistant to metabolic stress than beta cells in the T2D pancreas, because alpha cells express high levels of anti-apoptotic proteins [73]. On the other hand, in T1D, alpha cell mass seems to be decreased [74], indicating a complex inter-dependency between alpha and beta cells for development and functioning. Interestingly, the arrangement of endocrine cells within the human islet also hints at the developmental importance of paracrine signaling and coupling for the functional performance of these endocrine cells, as beta cells have multiple contacts with other endocrine cells within the islet [62,66]. Therefore, it is worth studying if beta cell replacement options under diabetic conditions that considered co-employing alpha cells and delta cells along with beta cells could better correct hyperglycemia [75]. To design a product that would include all major endocrine cells of the human islet, these cells could be purified individually using surface markers exclusive to each cell-type, followed by their aggregation in adequate percentages to represent a surrogate human islet, which can then be encapsulated and transplanted. However, if the cell-contact conferred functional properties of beta cells originate during fetal development of the pancreas, it may be more strategic to transplant progenitor cells that represent islet precursors and give rise to all desired endocrine populations assisting in glucose-responsiveness of beta cells in appropriate proportions, connected together as in the developing embryonic pancreas. However, the co-differentiation of ductal and acinar cell-types from transplanted pancreatic progenitors is an added disadvantage. Further studies on how to overcome this disadvantage are required. For example, the purification of multipotent pancreatic progenitors using markers, such as GP-2, prior to encapsulation is rather a convenient option to enhance the specification of beta cells [22,34]. Nevertheless, manipulating physiological conditions may also favor endocrine differentiation, as in the case of macro-encapsulation devices that are designed to provide a high oxygen pressure to the transplanted cells to enhance cell viability. Importantly, oxygen is a crucial factor for specification of endocrine progenitors as a high oxygen pressure was shown inhibit Notch signaling [76] which would in turn promote NGN3 expression at the expense of ductal specification [77]. Beta-O2 devices that have been used for islet transplantation have a replenishable gas chamber that constantly stipulate the transplanted cells with oxygen (https://beta-o2.com/efficient-oxygenation/). Improvements in the encapsulation system that would promote vascularization and engraftment to enhance oxygen supply are discussed later. In the case of no encapsulation, the selection of a transplantation site that has a higher oxygen pressure or is highly vascularized is rather preferable [78]. Alternatively, beta cell differentiation protocols could be adapted to generate a pancreatic islet organoid in vitro, consisting of all endocrine cell-types proportionately, self-organized into an islet-like structure, that can be directly transplanted as a terminally differentiated surrogate islet. This may be achieved by the purification of endocrine progenitors specifically, or by using a modified cell line that cannot generate non-pancreatic endocrine cells.

### 4.2. Role of Heterogeneous Beta Cell Subtypes in Glucose Stimulated Insulin Secretion

During development, islet cells exhibit heterogeneity, which may suggest the presence of different pancreatic progenitor populations. Also, heterogeneity may be developed in the postnatal period due to alterations in islet architecture, epigenome or beta cell plasticity [79]. However, pancreatic beta cell heterogeneity is starting to be uncovered as multiple sub-populations of beta cells have been identified. Beta cell heterogeneity was primarily illustrated with respect to glucose-responsiveness, including insulin biosynthesis and secretion [80,81,82]. Beta cell heterogeneity may arise due to dissimilar insulin content [60,83], or due to differentially active stimulation coupling and sensing machinery, such as the ion channels in the gap junctions [84]. These heterogeneous subpopulations have also been shown due to the differential expression of planar cell polarity marker Flattop (FLTP), whose higher expression was correlated with higher maturation and metabolic markers [85]. Beta cell hubs were shown to possess pacemaker properties for initiation of electrical stimulation in an intact islet [60,84]. It is interesting to note that these ‘hub’ subpopulations express lower levels of beta cell maturation markers such as PDX1, NKX6.1, and INS, but possess enhanced stimulatory properties [60], reflecting the functional relevance of immature beta cell populations. Additionally, the existence of multiple beta cell subpopulations have been indicated based on the differential expression of cell surface antigens and transcription factors [79]. However, their specific correlation to different physiological functions is yet to be fully elucidated. The identification of surface markers for functionally relevant beta cell sub-populations is required for their isolation and inclusion in the candidate product for cell therapy.

Importantly, the interaction of these heterogeneous beta cell subpopulations within the pancreatic islet has functional relevance in maintaining glucose homeostasis [86]. Specifically, beta cell communication through connexins, gap junctions and EphA-ephrin A signaling is crucial for controlling insulin levels. While loss of gap junctions did not affect the proportion of insulin secreted, it was shown to rather dysregulate synchronization of calcium currents between the connected beta cells [57,87]. The loss of connexin 36 specifically has been shown to result in an elevated levels of insulin release under basal glucose levels [88]. Studies have shown that EphA signaling in pancreatic beta cells is bidirectional, where the forward signaling is required for the inhibition of insulin secretion and reverse signaling stimulates insulin secretion [89]. Differential glucose sensitivity and electrical coupling is related to differentially expressed proteins in these beta cell subpopulations. There is evidence that these beta cell subpopulations characteristically sub-localize during islet development and inter-connect strategically, whereby they are conferred specific functional properties [90]. To be able to incorporate these in our candidate product, it is important to first determine if our in vitro beta cell directed differentiation protocols for hPSCs generate these beta cell subpopulations that may have differential functional properties and to optimize methods that can maximize the generation of a functionally relevant subpopulation. 

Achieving the right pancreatic organoid structure in vitro during differentiation process may aid in obtaining the right beta cell heterogeneity. In vivo maturation of pancreatic progenitors may allow this within the encapsulation device. However, terminally differentiated beta cells generated in vitro using optimized protocols may not have had the access to the in vivo micro-environmental factors necessary for appropriate islet organoid development, and as a result may have generated pancreatic beta cell sub-populations disproportionately.

Of note, beta cell heterogeneity has been shown to have disease relevance owing to phenomena such as beta cell neogenesis and de-differentiation. In T2D, multiple studies have revealed an alteration of beta cell identity due to the degranulation of insulin granules, and have hinted at a possible conversion of beta cell mass into a progenitor state or other endocrine cell types (see review [91]). This plasticity of the pancreatic islet leads to multiple subpopulations of beta cells, some of which may play a necessary role in adapting to cellular stress.

Therefore, it is plausible that allowing pancreatic progenitor cells to mature into hormone-secreting cells within the human body, exposed to the appropriate microenvironment, immune-isolated in an encapsulation device or semi-permeable membrane, would generate the required endocrine heterogeneity with the desired connections for optimum pancreatic endocrine function in diabetic conditions. However, extensive analysis upon the completion of a clinical trial for hPSC-derived pancreatic progenitors in T1D patients is needed to draw further conclusions. 

## 5. Encapsulation of hPSC-Derived Pancreatic Progeny for Cell Therapy

The in vivo maturation of hPSC-derived pancreatic progenitors or pancreatic beta cells indispensably requires a suitable transplantation site, as well as an appropriate encapsulation material or device. The pancreas provides an appropriate microenvironment for the maturation of islets. However, a surgical method for delivery and retreivability has limited its consideration as a candidate for transplantation site. One study transplanted islets in rats and found that normoglycemia is achieved with fewer islets, compared to the extra-pancreatic sites, such as the liver and the kidney [92]. However, it is worth noting that pancreatic progenitors transplanted subcutaneously or under kidney capsules have, nevertheless, resulted in their differentiation into functional beta cells [15,30], despite not being exposed to a ‘pancreatic’ microenvironment. For example, a previous study investigated the effect of the transplantation site on the generation of monohormonal insulin-secreting cells and found that the transplantation of pancreatic progenitors under mammary fat pads or kidney capsules do not affect their maturation into beta cells [15]. It is plausible that the crucial vascular system supplying nutrients and oxygen to these transplanted progenitors carries cues sufficient to facilitate their maturation into beta cells. 

An encapsulation device is, therefore, a crucial factor affecting the performance of the transplanted cells and it can control teratoma formation if the transplanted cells are contaminated with undifferentiated cells. Encapsulation of the transplanted pancreatic lineage is crucial in case of T1D to prevent an autoimmune reaction against the transplanted cells. Additionally, if the cell therapy product is from an allogenic source, for example, commercialized off-the-shelf hPSC-derived beta cells, then encapsulation is still a requirement. However, in the case of T2D or monogenic diabetes, there is a possibility of transplanting pancreatic progeny derived from the patient’s own cells, that circumvents the need for encapsulation [5]. 

Achieving the right design of an encapsulation device requires putting together variables, such as the biocompatibility properties of the membrane, exposure to the blood stream, and availability of nutrition and oxygen for the encapsulated cells amongst others [93]. Studies are being done on modification of the available materials to improve these properties of the biomaterial, and have mainly been developed for islet transplantation, both in macro- and micro-encapsulation systems. The assessment of islet function and viability following their coating with alginate derivatives is being widely investigated for improving islet transplantation outcomes. Purified alginate improved the survival of encapsulated islets and had a moderate effect on necrosis compared to non-purified alginate capsules [94,95]. Furthermore, certain alginate modifications are particularly interesting to study as they could circumvent immune response following transplantation of allogenic islets. Modification of alginate capsules using triazole-derivatives showed positive results in preventing immune cell activation at capsule surfaces in mice and non-human primates [96]. The incorporation of the chemokine CXCL12 in the alginate capsule protected the islets and improved their function by serving as an immune-isolating material without the need for immune-suppression [97]. Likewise, such alginate-based microencapsulation methods are now being applied for stem cell therapy, such as for hPSC-derived beta cells. CXCL12 coating was recently shown to prolong the viability of hPSC-derived beta cells in immune-competent mice without requiring immunosuppression by preventing fibrotic overgrowth [98]. In addition, the CXCL12 coating enhanced beta cell function by improving their glucose responsiveness, thereby making it an important biomaterial to study further for beta cell encapsulation.

While alginate-based microencapsulation is a promising option for beta cell replacement therapy, complete retrieval of the implanted microencapsulated islets in the portal vein, under kidney capsule or in the peritoneal space is challenging, often being invasive and incomplete [3,99,100]. Therefore, their use as an encapsulant for hPSC-derived beta cells, without a protective retrievable device could raise concerns as hPSC-derived beta cells have not yet been proven to be completely identical to pancreatic adult islets [27] and require their retrieval periodically to assess their viability, function, and to detect any teratomas formed. While studies on the formation of teratoma by unsorted terminally differentiated hPSC cultures containing target cells and uncommitted progenitors have revealed their expected frequency [101], it is unclear how the variability in differentiation efficiencies by different cell lines for the generation of beta cells would affect teratoma formation in vivo. However, the purification of fully mature hPSC-derived beta cells using surface markers prior to their microencapsulation could nullify some of these concerns. Therefore, further studies are required for the identification of these markers and their application. 

On the other hand, minimally invasive macro-encapsulation devices have also shown progress in delivering cell therapy products. The bilaminar TheraCyte macro-encapsulation device, which has an outer surface that promotes tissue engraftment or infiltration of blood vessels thus providing a close proximity of vasculature to the implanted cells and an inner membrane that prevents immune cell diffusion, has been shown to protect against immune-rejection of the transplanted pancreatic tissues in rodents. In addition, the device has allowed maturation of the hPSC-derived pancreatic lineages into pancreatic beta cells in animal models and proven competent at controlling hyperglycemia [9,29,31,102,103,104,105,106]. ViaCyte, leading the only clinical trial using hPSC-derived product for T1D treatment has reviewed the performance of its macro-encapsulation device, PEC-Encap, a TheraCyte adaptation, in the first phase of their study. The PEC-Encap is a combination of the hPSC-derived pancreatic progenitors enclosed in a semi-permeable, Encaptra drug delivery system, which was transplanted under the skin of T1D patients (https://viacyte.com/products/pec-encap-vc-01). The preliminary results indicated that the device is not effective in allowing its engraftment into the host tissue, which may impair supply of oxygen and nutrients to the therapeutic cells within the device (https://viacyte.com/products/pec-encap-vc-01). However, it was shown to protect the cells from immune invasion. To overcome these limitations, another ViaCyte system called the PEC-Direct, with a modified encapsulation membrane, is being currently employed in another clinical trial to assess its safety T1D hypoglycemic patients, however it requires immune-suppression (https://viacyte.com/products/pec-direct). Nonetheless, novel cell encapsulation systems are being developed, such as the Cell Pouch by Sernova (https://www.sernova.com/technology/) that facilitates formation of a pre-vascularized scaffold at the target site before cell delivery, that can advance cell therapy [107]. Another innovative device that aims to enhance oxygen availability to the encapsulated cells is the Beta-O_2_ device that consists of a gas chamber next to the encapsulated cells that allows the diffusion of oxygen to the cells (https://beta-o2.com/living-with-sair/). This chamber can be refilled occasionally to maintain a continuous supply of oxygen.

## 6. Immune Modulation and Suicide Gene Strategies for Improving the Safety of Beta Cell Therapy

Although massive progress has been achieved in the area of encapsulation technology, the engraftment of transplanted cells still presents challenges. Riddance of the encapsulation system altogether would certainly expose the graft to destruction by the immune system. However, hPSCs provide a remarkable platform for genome editing, whereby HLA antigens that provoke an immune response could be precisely eliminated to generate a universal donor hPSC line (Figure 4). Interestingly, while hPSC-derived pancreatic progenitors presented low levels of HLA antigens, their expression was upregulated upon in vivo maturation to beta cells [37]. A higher expression of HLA on hPSC-derived beta cells therefore makes them a target for destruction by the immune cells. Recently, it was shown that the deletion of either beta-2-microglobulin, that abolishes the expression of any class I HLA protein or simply HLA-A and HLA-B, but only one allele of HLA-C allowed the hPSC graft to evade T cell attack [108]. While the deletion of all HLA proteins would completely prevent antigen presentation in case of infections, retaining HLA-C and minor HLA proteins would permit this while still providing hypo-immunogenicity for the graft (Figure 4). Other approaches for immune-cloaking have been explored, such as the beta cell-specific overexpression of PDL1-CTLA4Ig molecules that prevented T cell activation and facilitated escape from the immune system, and in turn preserved the beta cell mass and prevented diabetes [109]. Therefore, immune modulation strategies could prove helpful in combatting challenges related to graft rejection and engraftment. 

While hPSC-derived pancreatic progeny holds great promise for treating diabetes, there are certain undeniable concerns regarding their clinical application. One such concern is the uncontrolled growth of uncommitted progenitor cells or de-differentiation of mature endocrine cells upon transplantation. There has been remarkable progress in this area that tackles the possibility of teratoma formation by the transplanted cells, one such being ‘suicide switches’. A suicide switch is a gene whose expression causes cell death, which can be induced in vivo to kill the unwanted grafted cells [110]. For example, an inducible Caspase 9 gene cassette is introduced using a lentiviral vector in hiPSCs with EF1a promoter that, when exposed to drugs that activate the Caspase 9 gene in vivo following transplantation, resulted in eradication of the teratoma formed by the modified hiPSCs [111]. However, there is a likelihood of this approach to destroy the differentiated therapeutic cells along with the malicious ones upon exposure to the drug. Other genes, such as herpes simplex virus thymidine kinase (HSV-tk), introduced with the promoter of a variety of markers characterizing pluripotency, such as Telomerase (hTERT) [112], cell cycle regulator CDK1 [113], as well as microRNA expressed in undifferentiated cells such as Let7 [114], have also shown encouraging results and are being further investigated in clinical studies. HSV-tk renders the modified cells sensitive to specific drug in vivo that results in the loss of progenitor-like or proliferative cells. However, it has been shown that the modified cells could develop resistance to the drug in vivo, thus making their complete removal uncertain [114,115]. The HSV-tk/GCV is exciting, however more recently, an innovative double suicide gene approach was developed. This approach introduced the HSV-tk under the telomerase promoter and another gene, nitroreductase (NTR), that is constitutively active, conferred sensitivity to a second drug. Importantly, NTR was flanked by LoxP and Cre was targeted to the INS promoter, whereby INS expression would result in the excision of NTR gene. Therefore, on exposure to the two drugs combined, the undifferentiated cells would undergo apoptosis in vivo and the terminally differentiated cells that do not express INS (non-beta cells) would be susceptible to apoptosis. This approach not only protects from teratoma incidence, but also enriches for insulin-secreting cells, thereby enhancing the impact of beta cell therapy [112]. While the group found very rare GCG-expressing cells following drug exposure, polyhormonal cells that co-expressed INS with other hormones including GCG were also enriched, along with monohormonal beta cells [112].

## 7. Concluding Remarks and Future Perspectives

Stem cell-derived cell therapy is indeed a very promising option for diabetes treatment. The improvement of in vitro beta cell differentiation protocols is required for their scalable generation for therapeutic needs. However, studies on islet development have hinted at the importance of intra-islet communication and paracrine signaling that play significant roles in potentiating adequate insulin secretion responses by pancreatic beta cells. Achieving the appropriate islet architecture in vitro using innovative methods could enhance hPSC-derived beta cell functionality. Additionally, since beta cell heterogeneity exists in normal human islet, it may be of interest to establish similar heterogeneity in vitro. Identifying beta cell phenotypes, specific markers for each cell subpopulation, and the mechanisms underlying their regulation will advance our understanding of beta cell function. Whether in vivo maturation of pancreatic progenitors could facilitate co-generation of these functionally relevant beta cell subpopulations needs to be examined. Furthermore, studying how this beta cell heterogeneity is altered in diabetic conditions will allow targeting of the affected subpopulations in the candidate product for more robust glycemic control. 

For the purposes of developing an off-the-shelf product, hPSC-derived pancreatic progenitor cells have shown the capacity to be expanded and cryopreserved that facilitates scalability [38,39]; however, hPSC-derived beta cells are yet to be evaluated. While these pancreatic progenitors could not retain NKX6.1 expression in PDX1+/SOX9+ expanded cells, they were nevertheless able to generate C-peptide expressing cells upon employing appropriate differentiation protocol, as well as ductal cells [38,39]. Therefore, it is likely that the variability across cell lines and differentiation protocols would add to the alterations in the identity of cryopreserved cells post-thawing, which can be overcome by isolating a pure population of beta cell precursors, for example, by selecting for GP2. However, the evaluation of whether GP2-purified cells could be cryopreserved and expanded, and whether these cryopreserved cells could differentiate into functional beta cells, at least in vivo, is required prior to facilitating their cell banking for transplantation therapy. 

Given the indispensable role of other pancreatic endocrine cells such as alpha and delta, and functionally dissimilar heterogenous beta sub-populations, it is key to take them into account while composing the ideal cell therapy candidate for diabetes. Whether the functional benefits from these populations are conferred or attained during pancreatic development, or are dependent only on their proximity during elevation in glucose levels to insulin-secreting cells, would determine if they need to be co-generated with beta cells in vitro to form appropriate contacts, or if they can be isolated individually using exclusive surface markers and put together prior to transplantation. Finally, if they originate from a common progenitor, it may be rather efficient to transplant the progenitor cells and allow them to mature in vivo into functional endocrine cells, exposed to the right microenvironment and attaining the desired architecture. In line with this concept, hESC-derived pancreatic progenitors transplanted without sorting differentiate in vivo into an islet-like structure expressing different pancreatic hormones (INS+, GCG+, and SST+) [9]. While recent reports demonstrated the generation of hPSC-derived beta cells showing some similarities to human beta cells, challenges related to functional characteristics and cell line efficiency variations remain. These variations are less evident in hPSC-derived pancreatic progenitors, which are currently employed in clinical trials for T1D patients.

Optimizing an encapsulation device suited to the right cell therapy product is also an important factor in attaining successful results for the long-term reversal of hyperglycemia. Efforts should be aimed at improving engraftment of the transplanted cells and maximizing nutrient availability within the graft. Incorporating lessons learnt from islet transplantation studies about site pre-vascularization and encapsulation technology could help to enhance the success of hPSC-derived cell therapy for diabetes. 

Genome editing of hPSCs has indeed brought us to an era of ‘off-the shelf’ beta cell therapy for diabetes. Immune modulation of hPSCs either by deletion of HLA antigens, immune-cloaking approaches, incorporation of suicide genes or their combination and other optimized gene editing strategies could allow generation of universal donor cell lines that would greatly facilitate wide-scale clinical application of hPSC-derived pancreatic progeny. Given the crucial role played by the endocrine cells within the islet and the cell–cell contacts formed in controlling glycemic levels, a cell therapy candidate that either resembles or could give rise to a surrogate islet would potentially be the closest to an endogenous scenario. hPSC-derived pancreatic progenitors could mature in vivo to beta cells recapitulating islet development, thereby establishing the cell-cell contacts necessary for their functionality and glucose responsiveness. Any non-islet cells generated in vivo, such as ductal and acinar cells, or non-committed progenitors, could be eliminated by the activation of the suicide gene and terminally differentiated endocrine cells of the islet could be enriched using the double suicide gene strategy that selects endocrine cells. Immune-modulated cell lines would eliminate the need for encapsulation, which could promote the vascularization of the graft more efficiently, with a careful selection of the transplantation site. While this strategy could promote a wide-scale hPSC-derived beta cell therapy, extensive work in the area is still warranted to ensure patient safety and graft functionality.

Finally, the cost-effectiveness of beta cell replacement therapy using hPSC-derived pancreatic progeny is a crucial factor shaping its clinical translation. A combination of small molecules, recombinant proteins and growth factors, as well as culture medium are required for making beta cells from hPSCs and are expensive in addition to the cost for labor and expertise. While a typical islet transplantation procedure would account for the costs of acquiring the organ, islet isolation, failed isolation attempts, surgical procedures, and multiple islet infusions, a direct analysis of how it would compare to a cell therapy treatment using hPSCs is needed. Nevertheless, any cost grievances in using hPSCs for cell therapy are trumped by their potential for expansion and scalability. This leverages their use over adult islets, which are limited by donor availability.

## Figures and Tables

**Figure 1 cells-09-00283-f001:**
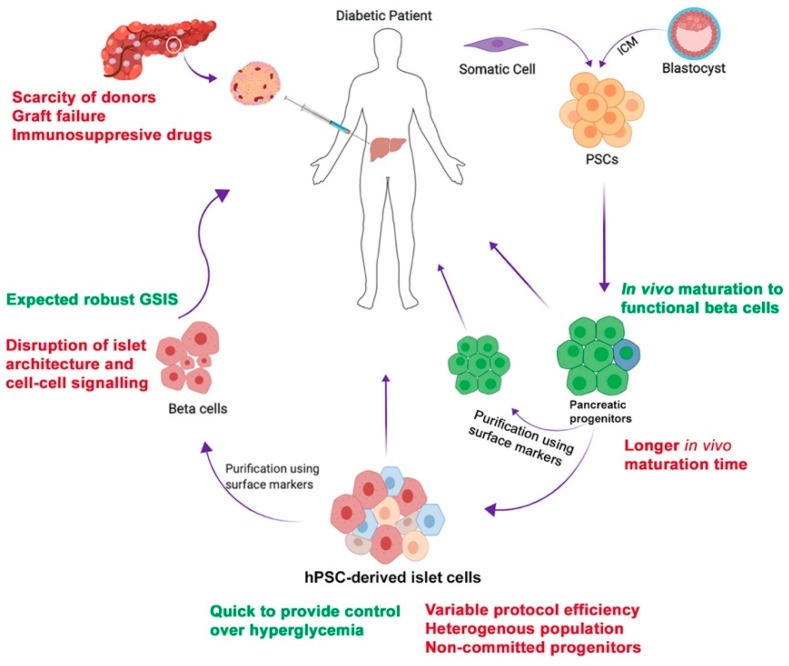
Schematic representation showing the potential use of human pluripotent stem cell (hPSC) for diabetes treatment. hPSCs derived from the inner cell mass (ICM) of the blastocyst and hiPSCs generated from patient somatic cells can be differentiated into pancreatic progenitors that mature in vivo into glucose-responsive beta cells following transplantation. These pancreatic progenitors can be purified as well as encapsulated prior to transplantation. Alternatively, hPSC-derived pancreatic progenitors can be differentiated into pancreatic beta cells in vitro and then transplanted in diabetic patient. In vitro differentiation to beta cells yields non-committed progenitors or polyhormonal and other endocrine cells. Therefore, these hPSC-derived beta cells can be purified using specific cell surface markers, that could disrupt the islet architecture recapitulated during differentiation, that may result in loss of cellular contact-conferred functional properties.

**Figure 2 cells-09-00283-f002:**
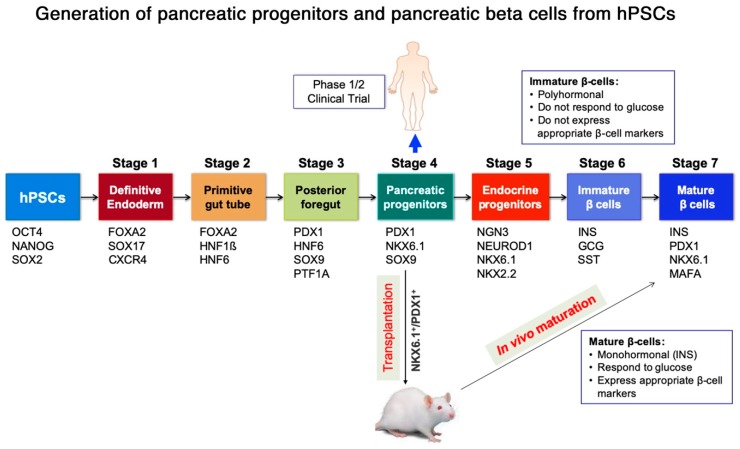
Different stages of pancreatic beta cell development during the differentiation process. According to Rezania et al. protocol the differentiation of hPSCs into pancreatic beta cells occurs though their differentiation into seven stages, which are confirmed by examining stage-specific markers [8]. Stage 4 (pancreatic progenitors) co-expressing PDX1 and NKX6.1 (PDX1^+^/NKX6.1^+^) is currently used in clinical trial for diabetes treatment. PDX1+/NKX6.1+ cells transplanted into mouse model can differentiate in vivo into mature insulin-secreting cells. Furthermore, mature beta cell stage generated in vitro can be directly transplanted into mouse model.

**Figure 3 cells-09-00283-f003:**
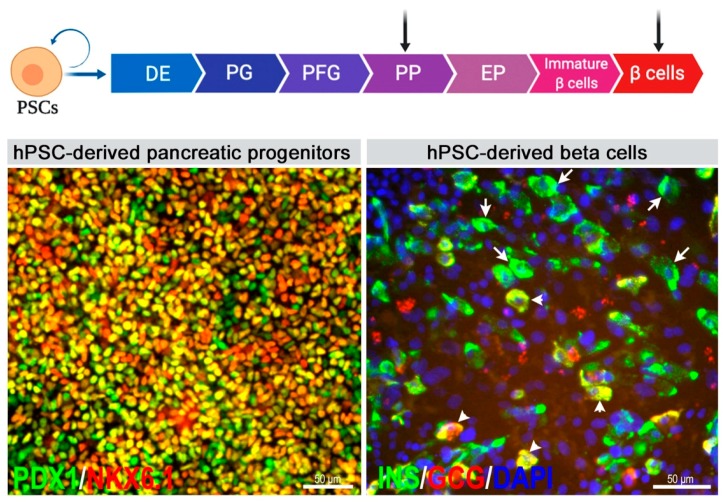
Directed differentiation of human pluripotent stem cells (hPSCs) into pancreatic progenitors and beta cells. Left panel, pancreatic progenitors co-expressing the transcription factors, PDX1 and NKX6.1, are considered bona fide beta cell precursors and can be differentiated with a high efficiency from optimized in vitro protocols [21]. Right panel, hPSC-derived pancreatic beta cells expressing INS alone (monohormonal) (arrows) or INS and GCG (polyhormonal) (arrowheads) (Abdelalim’s lab, unpublished data). PSCs, Pluripotent stem cells; DE, Definitive endoderm; PG, Posterior gut tube; PFG, Posterior foregut; PP, Pancreatic progenitors; EP, Endocrine progenitors; PDX1, Pancreatic and duodenal homeobox 1; NKX6.1, NK6 homeobox 1; INS, INSULIN; GCG, GLUCAGON.

**Figure 4 cells-09-00283-f004:**
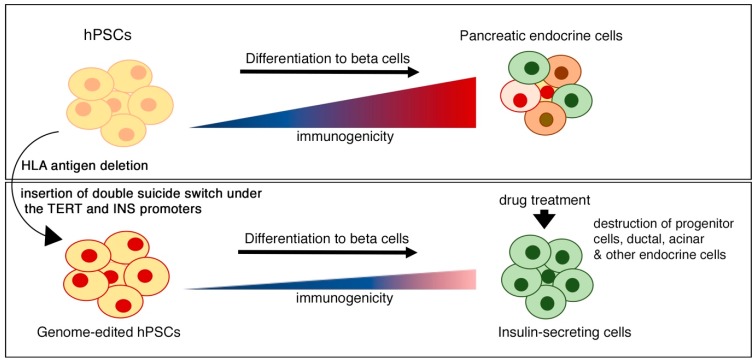
Immuno-modulation strategies for ‘off-the-shelf’ clinical use of human pluripotent stem cell-derived beta cells. A universal product can be developed by combining gene editing for different strategies, such as HLA antigen disruption to reduce immunogenicity and insertion of double suicide switches to eliminate proliferative, non-committed progenitors as well as other pancreatic lineages, thereby enriching insulin-secreting beta cells.

**Table 1 cells-09-00283-t001:** Comparison between pancreatic progenitors and beta cells derived from human pluripotent stem cell (hPSCs).

Feature	Pancreatic Progenitors	Pancreatic Beta Cells
Key transcription factors	PDX1, NKX6.1, and FOXA2 [7]	NKX6.1, MAFA, and PDX1 [7]
Surface markers	CD24 [32], CD142 [33], GP2 [22,34]	CD49a [35]
Duration of in vitro differentiation protocol	~2 weeks [15,21]	~30–36 days [17,18,19,36]
Method of differentiation	Monolayer [15,21] and aggregation in suspension [24,25]	Aggregation in suspension [17,18,19]
Display of human c-peptide secretion following transplantation in vivo (rodents)	~3–4.5 months [9,22]	~3–14 days [17,18,19]
After transplantation	Islet-like structure (INS+, GCG+, and SST+ cells)	INS+ cells
Clinical trials	Yes	No
HLA expression	Low [37]	High [37]
Off-target differentiation	Acinar and ductal cells could be co-generated [15,33]	Polyhormonal and other endocrine cells [8,17,19]
Generation of functionally relevant heterogenous beta cell subpopulations	undetermined	undetermined
Expansion and freeze-thaw potential for storage	Yes [38,39]	undetermined

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
