# Peer review of "Stem Cell Therapy for Diabetes: Beta Cells versus Pancreatic Progenitors"

_cells, 2020, doi:10.3390/cells9020283_

Round 1

Reviewer 1 Report

This is a very well written review discussing the advantages and challenges associated with the use of the human pluripotent stem cells (hPSCs)-derived pancreatic progenitor or beta cells for diabetes cell therapy. Furthermore, the manuscript address co-generation of functionally relevant islet cell subpopulations and structural properties contributing to glucose responsiveness of beta cells, as well as the available encapsulation technology for these cells. The literature review support is very good and updates the most relevant achievements in related research. The pros and cons of using beta cells versus pancreatic progenitors as stem cell therapy for diabetes have been nicely covered.

Minor suggestions:

In paragraph 6, it could be very useful to include an additional figure explaining the suicide gene strategies.

Author Response

We thank the reviewer for his insight. Following the reviewer's suggestion, we have prepared a new Figure (Figure 4). Figure 4 and its legend were added to the manuscript.

Figure 4: Immuno-modulation strategies for ‘off-the-shelf’ clinical use of human pluripotent stem cell-derived beta cells. A universal product can be developed by combining gene editing for different strategies such as HLA antigen disruption to reduce immunogenicity and insertion of double suicide switches to eliminate proliferative, non-committed progenitors as well as other pancreatic lineages, thereby enriching insulin-expressing cells.

Reviewer 2 Report

This is an interesting review which discusses the possible advantages and challenges of using hPSC-derived beta cells or hPSC-derived pancreatic progenitors for the replacement of the destroyed pancreatic beta cells in diabetes (generally in type 1 diabetes). Although islet transplantation from donors is currently a more common treatment method, it has several limitations. Hence, using hPSC-derived beta cells or hPSC-derived pancreatic progenitors for the replacement of the destroyed pancreatic beta cells seems to be a better option in the future. I only have some minor comments for this review as below:

In line 53 and later parts of the manuscript, the authors states that “one clinical trial has been approved for hPSC-derived 52 product for T1D treatment, led by the company ViaCyte”. Please provide the clinical trial registration number (from https://clinicaltrials.gov/ or other clinical trial registries) in the review in order for the readers to refer to the details and updates of the clinical trial. Between using hPSC-derived beta cells and hPSC-derived pancreatic progenitors for treatment of type 1 diabetes, which of methods has a cost advantage? What about their cost effectiveness in comparison to the current more commonly used method - islet transplantation from donors? I suggest that it will be good to have a short discussion of the cost issues in the review since cost is also an important factor to consider in actual clinical practices. In line 340 – 342, the sentence “This section is not mandatory, but can be added to the manuscript if the discussion is unusually long or complex” seems to be a sentence from the journal template or etc and should be deleted from the review. Minor language and formatting mistakes: line 147 etc “in vivo” should be italic, line 130 etc “in vitro” should be italic, etc. Proofreading of the manuscript for language and formatting mistakes is recommended.

Author Response

Q1. In line 53 and later parts of the manuscript, the authors states that “one clinical trial has been approved for hPSC-derived 52 product for T1D treatment, led by the company ViaCyte”. Please provide the clinical trial registration number (from https://clinicaltrials.gov/ or other clinical trial registries) in the review in order for the readers to refer to the details and updates of the clinical trial.

We have added the clinical trial registration numbers for ViaCyte in line 62:

ClinicalTrials.gov Identifier: NCT03163511, NCT02239354

Q2. Between using hPSC-derived beta cells and hPSC-derived pancreatic progenitors for treatment of type 1 diabetes, which of methods has a cost advantage? What about their cost effectiveness in comparison to the current more commonly used method - islet transplantation from donors? I suggest that it will be good to have a short discussion of the cost issues in the review since cost is also an important factor to consider in actual clinical practices.

The reviewer highlighted an important point to be considered for developing an off-the-shelf product. There is no cost analysis available yet for a cell therapy treatment with hPSC-derived candidate; however, we have weighed in factors affecting cost estimations for both hPSCs and islets at the end of section 7 (Concluding remarks and future perspectives) as follows:

 “Finally, cost-effectiveness of a beta cell replacement therapy using hPSC-derived pancreatic progeny is a crucial factor shaping its road to the clinic. A combination of small molecules, recombinant proteins and growth factors as well as culture medium are required for making beta cells from hPSCs that are expensive in addition to the cost for labor and expertise. While a typical islet transplantation procedure would account for costs of acquiring the organ, islet isolation, failed isolation attempts, surgical procedures and multiple islet infusions; a direct analysis of how it would compare to a cell therapy treatment using hPSCs is needed. Nevertheless, any cost grievances in using hPSCs for cell therapy is trumped by their potential for expansion and scalability. This leverages their use over adult islets which are limited by donor availability.”

Q3. In line 340 – 342, the sentence “This section is not mandatory, but can be added to the manuscript if the discussion is unusually long or complex” seems to be a sentence from the journal template or etc and should be deleted from the review.

Following the reviewer’s comment, we have deleted the sentence from the manuscript

Q4. Minor language and formatting mistakes: line 147 etc “in vivo” should be italic, line 130 etc “in vitro” should be italic, etc.

Following the reviewer’s comments, we modified “In vitro” and “In vivo” throughout the manuscript.

Q5. Proofreading of the manuscript for language and formatting mistakes is recommended. 

We re-checked and corrected the grammar throughout the manuscript.

Reviewer 3 Report

This review evaluates the relative merit of pancreas progenitor cells, compared with mature beta cells, both generated in vitro by differentiation of pluripotent human stem cells, for cell therapy of diabetes.

The topic is of clinical relevance. The review is comprehensive, clear, and well-written. There are a few minor points that need to be addressed:

The main argument against using mature beta cells is the variable results obtained with different pluripotent stem cell lines subjected to the same differentiation protocol. However, it is unclear to what extent this variability is also manifested at the pancreas progenitor stage, and among mature cells developing from these cells in vivo. It would be of interest to describe the potential of expanding pancreas progenitor cells in vitro, once they reached this stage, perhaps following selection, while preventing further differentiation, for cell banking prior to transplantation. In Fig. 2 it is not mentioned which specific protocol is shown in the scheme. Presumably this is the Rezania et al. protocol, as the ones from the Melton and Hebrok labs, to name just two, differ somewhat. Please specify and reference. Please use “beta” or “β” consistently throughout the text.

Author Response

Q1. The main argument against using mature beta cells is the variable results obtained with different pluripotent stem cell lines subjected to the same differentiation protocol. However, it is unclear to what extent this variability is also manifested at the pancreas progenitor stage, and among mature cells developing from these cells in vivo. It would be of interest to describe the potential of expanding pancreas progenitor cells in vitro, once they reached this stage, perhaps following selection, while preventing further differentiation, for cell banking prior to transplantation.

 We thank the reviewer for raising this crucial point. We have discussed this point in section 7 (7. Concluding remarks and future perspectives) as follows:

“For the purposes of developing an off-the-shelf product, hPSC-derived pancreatic progenitor cells have shown the capacity to be expanded and cryopreserved that facilitates scalability [116, 117]; however, hPSC-derived beta cells are yet to be evaluated. While these pancreatic progenitors could not retain NKX6.1 expression in PDX1+/SOX9+ expanded cells, they were nevertheless able to generate C-peptide expressing cells upon employing appropriate differentiation protocol, as well as ductal cells [116, 117]. Therefore, it is likely that the variability across cell lines and differentiation protocols would add to the alterations in the identity of cryopreserved cells post-thawing, it can be overcome by isolating a pure population of beta cell precursors, for example, by selecting for GP2. However, evaluation of whether GP2-purified cells could be cryopreserved and expanded and whether these cryopreserved cells could differentiate into functional beta cells, at least in vivo, is required prior to facilitating their cell banking for transplantation therapy.”

Also, we have added two references in the table (Ref # 116, 117: Trott et al, Stem Cell Reports, 2017; Konagaya et al, Scientific Reports, 2019) in the feature “Expansion and freeze-thaw potential for storage”.

Q2. In Fig. 2 it is not mentioned which specific protocol is shown in the scheme. Presumably this is the Rezania et al. protocol, as the ones from the Melton and Hebrok labs, to name just two, differ somewhat. Please specify and reference. Please use “beta” or “β” consistently throughout the text. 

 Following the reviewer’s comment, we have specified in the figure legend that this protocol according to Rezania et al and we added the reference in the legend (Rezania et al., 2014). Also, we used “beta” throughout the text.
